# PPMStereo: Pick-and-Play Memory Construction for Consistent Dynamic Stereo Matching

**Yun Wang[1], Junjie Hu[2],* Qiaole Dong[3],† Yongjian Zhang[4]**
**Yanwei Fu[3], Tin Lun Lam[2], Dapeng Wu[1]**
[1]City University of Hong Kong, [2]The Chinese University of Hong Kong, Shenzhen
[3] Fudan University,[4] Shenzhen Campus, Sun Yat-sen University
ywang3875-c@my.cityu.edu.hk, dpwu@ieee.org,
{qldong18, yanweifu}@fudan.edu.cn
zhangyj85@mail2.sysu.edu.cn,
{hujunjie,tllam}@cuhk.edu.cn

## Abstract

Temporally consistent depth estimation from stereo video is critical for real-world applications such as augmented reality, where inconsistent depth estimation disrupts the immersion of users. Despite its importance, this task remains challenging due to the difficulty in modeling long-term temporal consistency in a computationally efficient manner. Previous methods attempt to address this by aggregating spatio-temporal information but face a fundamental trade-off: limited temporal modeling provides only modest gains, whereas capturing long-range dependencies significantly increases computational cost. To address this limitation, we introduce a memory buffer for modeling long-range spatio-temporal consistency while achieving efficient dynamic stereo matching. Inspired by the two-stage decision-making process in humans, we propose a **P**ick-and-**P**lay **M**emory (PPM) construction module for dynamic **Stereo** matching, dubbed as **PPMStereo**. PPM consists of a 'pick' process that identifies the most relevant frames and a 'play' process that weights the selected frames adaptively for spatio-temporal aggregation. This two-stage collaborative process maintains a compact yet highly informative memory buffer while achieving temporally consistent information aggregation. Extensive experiments validate the effectiveness of PPMStereo, demonstrating state-of-the-art performance in both accuracy and temporal consistency. Codes are available at https://github.com/cocowy1/PPMStereo.

## 1   Introduction

Stereo matching refers to binocular disparity estimation, which is a fundamental computer vision task focused on estimating the disparity between a pair of rectified stereo images [53, 21, 34]. Deep learning-based stereo matching methods have achieved remarkable progress in terms of accuracy [53, 58, 50, 11], efficiency [46, 59, 52, 2], and robustness [42, 64, 63, 55]. Despite impressive performance for static scenes, these methods exhibit severe temporal inconsistencies when applied to dynamic scenes [24]. This manifests itself as flickering artifacts and blurred disparity maps due to the absence of effective inter-frame temporal information integration. Therefore, the algorithm deployment in dynamic scenarios such as autonomous driving, robotics, and augmented reality platforms is limited, which requires temporally consistent disparity maps.

---

*Corresponding author.
†Project Leader

39th Conference on Neural Information Processing Systems (NeurIPS 2025).

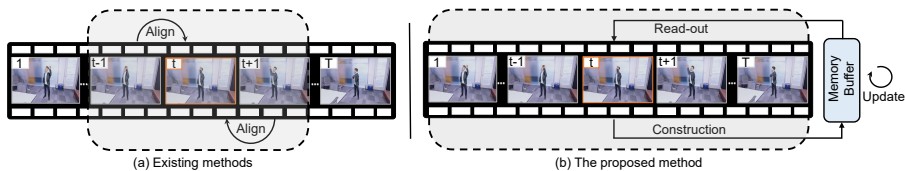

Figure 1: Comparison between prior methods (a) and our method (b). For the $t$-th frame, prior works process video sequences using small temporal sliding windows with attention or optical flow, restricting cost information propagation. Our method captures long-range spatio-temporal relationships across the input sequence by constructing and updating a compact memory buffer.

To address the task of dynamic stereo matching, recent approaches start to incorporate temporal cues from two main perspectives to achieve temporally consistent estimation. Some methods [30, 61, 12] refine the current disparity with disparity or motion of previous neighbor frame, while achieving limited improvements in temporal consistency due to the narrow temporal context. Secondly, other approaches [24, 22] (Fig. 1 (a)) expand the temporal receptive field by using attention mechanisms to model spatio-temporal relationships [24] within a sliding window while treating all frames equally, which overlooks variations in frame reliability. BiDAStereo [22] further depends on optical flow priors for alignment, may incurring errors from flow inaccuracies and high computational cost. Overall, video-based methods face a core trade-off: narrow context yields marginal improvements, whereas naively using all frames drives up computation without reliability awareness.

Naturally, these considerations lead to a key question: How can we design a model that effectively models long-range temporal relationships while maintaining computational efficiency? To answer this question, we draw inspiration from recent advances in sequence processing and bring a memory buffer into the dynamic stereo matching task. We present **P**ick-and-**P**lay **M**emory for dynamic **Stereo** matching named **PPMStereo** which enables effective and efficient utilization of reference frames for long-range spatio-temporal modeling by dynamically reducing redundant frames while selectively retaining and leveraging the most valuable frames throughout the video sequence to ensure accuracy and efficiency, as illustrated in Fig. 1 (b).

Specifically, our method draws inspiration from human decision-making in complex scenarios, which typically involves the 'pick' process that identifies the most essential elements from a set of candidates and the 'play' process that meticulously balances and leverages the identified elements [5, 18, 38]. In this paper, we propose a novel Pick-and-Play Memory construction method for video stereo matching. Specifically, the 'pick' process identifies the most relevant $K$ frames from $T$ reference frames for the current frame. To facilitate this process, we introduce a novel Quality Assessment Module (QAM), which evaluates each frame's contribution by jointly evaluating confidence, redundancy, and similarity of reference frames. Upon identifying the most relevant $K$ frames, the 'play' process adaptively weights the importance of the features extracted from those $K$ selected frames via a dynamic memory modulation mechanism. Subsequently, we utilize an attention-based memory read-out mechanism that queries the high-quality memory buffer using the current frame's contextual feature, yielding temporally and spatially aggregated cost features. By combining this aggregated cost feature with the current cost and context features, we can use GRU modules to regress the residual disparities.

Extensive experiments show that our method achieves state-of-the-art temporal consistency and accuracy. Specifically, on both the clean and final pass of the Sintel [6] dataset, our model achieves a temporal end-of-point error (TEPE) of 0.62 and 1.11 pixels, with 3-pixel error rates of 5.19% and 7.64%, respectively. Compared to the previous SoTA method, BiDAStereo [22], this represents a 17.3% and 9.02% reduction in TEPE and a 9.74% and 10.32% improvement in 3-pixel error rate, while enjoying lower computational costs. Overall, the contributions of our work can be summarized as follows: (1) We introduce PPMStereo, the first work that successfully builds a memory buffer to tackle dynamic stereo matching, allowing for long-range spatio-temporal modeling in a computationally efficient way. (2) We propose a novel 'Pick-and-Play' memory buffer construction method that first identifies the key subset of reference frames with the pick process and then effectively aggregates them with a play process, enabling highly accurate and temporally consistent disparity estimation. (3) Extensive experiments demonstrate that PPMStereo achieves state-of-the-art performance across multiple dynamic stereo matching benchmarks.

## 2 Related Work

**Deep Stereo Matching.** Existing deep stereo matching methods [47] primarily focus on cost volume aggregation for network and representation design. These approaches are generally categorized into regression-based [34, 25, 59, 62, 42, 56, 53] and iterative-based methods [32, 27, 54, 58, 50, 64]. Regression-based methods typically regress a probability volume to estimate disparity maps, which can be further divided into 2D [34, 31, 59, 54] and 3D cost aggregation approaches [19, 62, 42, 33, 43, 53, 52]. These methods either directly regress disparity across a predefined global range [25, 62, 53] or employ a coarse-to-fine refinement strategy to improve accuracy [42, 43, 33]. Recently, iterative-based methods [57, 32, 4, 54, 28, 50, 55, 51, 26] have emerged as the dominant paradigm in stereo matching. These methods leverage multi-level GRU or LSTM modules to iteratively refine disparity maps through recurrent cost volume retrieval, achieving state-of-the-art performance. However, despite their remarkable results, these approaches infer disparities independently for each frame, ignoring temporal correlations across video sequences. As a result, they often suffer from poor temporal consistency, which manifests as flickering artifacts in the disparity outputs.

**Dynamic Stereo Matching.** A few methods in stereo matching have focused on leveraging temporal cues from dynamic scenes to enhance disparity consistency. These methods can be mainly categorized into two paradigms: (i) **Adjacent-frame Integration**, which propagates disparity or motion fields from the immediately preceding frame to maintain local temporal smoothness. These works [30, 65, 12, 61] typically employ warped disparity or motion estimates for robust initialization, thereby enhancing the temporal consistency. However, these methods are limited by their reliance on only the most recent frame, resulting in a narrow temporal receptive field. (ii) **Multi-frame Integration**, which employs sliding-window aggregation across extended temporal contexts to enforce temporal consistency through attention mechanisms (DynamicStereo) [24] or optical flow priors (BiDAStereo) [22]. Despite their strengths, attention-based methods treat all frames equally without assessing the reliability of reference frames and suffer from high computational costs with a large window. Additionally, flow-based methods are sensitive to optical flow estimation errors and introduce extra computational overheads. In contrast, our method effectively aggregates long-range spatio-temporal information from a compact yet high-quality memory buffer. Thanks to our 'pick' process, PPMStereo remains computationally efficient, even with the enlarged temporal window.

**Memory Cues for Video Tasks.** Prior works have explored memory model [45] across various video tasks, including optical flow [15], segmentation [37, 66, 9, 10], tracking [60, 17], and video understanding [44, 20], demonstrating its significant effectiveness for video-related tasks. Among them, XMem [9] consolidates memory by selecting prototypes and evicting obsolete features via a least-frequently-used policy, while RMem [66] improves the segmentation accuracy by using a fixed frame memory bank [1]. Prior works have explored memory model [45] across various video tasks, including optical flow [15], segmentation [37, 66, 9, 10], and video understanding [44, 20], demonstrating its significant effectiveness for video-related tasks. Among them, XMem [9] consolidates memory by selecting prototypes and evicting obsolete features via a least-frequently-used policy, while RMem [66] improves the segmentation accuracy by using a fixed frame memory bank [1]. The closest related work is MemFlow [15], which develops an adjacent-frame memory buffer framework to aggregate spatio-temporal motion for optical flow estimation. While effective for optical flow, MemFlow yields limited gains when directly applied to dynamic stereo matching, as it only retains the immediate adjacent frame. Expanding its temporal scope without reliability assessment introduces redundant and noisy cues. In contrast, our method adaptively updates and modulates the most valuable memory cues across the entire sequence, enabling robust long-range spatio-temporal modeling while filtering out inferior ones, leading to significant performance improvements.

## 3 Methodology

### 3.1 Overview

Dynamic stereo matching seeks to recover a sequence of temporally consistent disparity maps $\{\mathbf{d}^t\}_{t\in(1,T)} \in \mathbb{R}^{H\times W}$ from stereo video frames $\{\mathbf{I}_L^t, \mathbf{I}_R^t\}_{t\in(1,T)} \in \mathbb{R}^{H\times W\times 3}$, where $T$ is the number of frames, $H$ and $W$ are the height and width dimensions. However, prior approaches struggle to capture long-range temporal dependencies without incurring prohibitive cost. To address this, we introduce **PPMStereo**, which augments the DynamicStereo backbone [24] with a Pick-

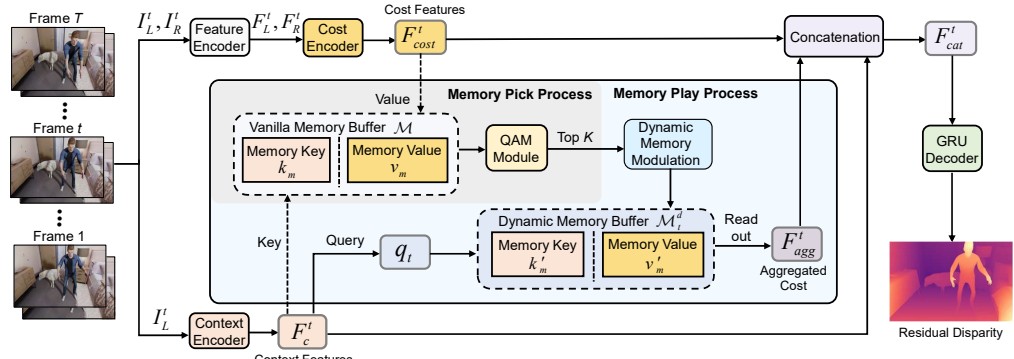

Figure 2: An overview of PPMStereo. The gray part is the memory 'pick' process, and the blue part is the memory play process. Our PPMStereo employs a dynamic memory buffer for modeling long-range spatio-temporal relationships while maintaining computational efficiency.

and-Play Memory (PPM) module that selectively aggregates high-quality references into a compact, query-adaptive buffer, thereby strengthening spatio-temporal modeling while remaining efficient. As illustrated in Fig. 2, the overall pipeline proceeds as follows: *(1) Feature Extraction:* a shared encoder extracts multi-scale features $\{F_L^t, F_R^t\}_{(s)} \in \mathbb{R}^{sH \times sW \times C}$ at scales $s \in \{1/16, 1/8, 1/4\}$, with $C$ channels. These pyramidal representations provide both receptive-field diversity and a convenient substrate for multi-scale matching. *(2) Cost Volume Construction:* at each time step $t$, we construct a 3D correlation volume from $\{F_L^t, F_R^t\}_{(s)}$ and pass it through a lightweight cost encoder to obtain matching costs $F_{cost}^t$, subsequently projected to a value embedding $v_t$. *(3) Context Encoding:* A context encoder operating on the left view produces $F_c^t$, which are linearly projected to a query $q_t$ and $k_t$. *(4) Memory Buffer Initialization and Update:* To expose the model to long-range spatio-temporal correlations, we initialize a vanilla memory $\mathcal{M} = \{k_m \in \mathbb{R}^{L \times C}, v_m \in \mathbb{R}^{L \times C}\}$ that stores $k_m = \{k_1, \ldots, k_T\}$ and $v_m = \{v_1, \ldots, v_T\}$ with $L = T \times sH \times sW$. This naive memory buffer stores all reference-frame features, making per-iteration queries prohibitively expensive. To retain accuracy without sacrificing efficiency, we introduce the *Pick-and-Play Memory (PPM)*: driven by a Quality Assessment module (omitting the iteration index $n$ for brevity), PPM first *picks* the most informative references to construct a compact, dynamic buffer $\mathcal{M}_t^d = \{k_m' \in \mathbb{R}^{L' \times C}, v_m' \in \mathbb{R}^{L' \times C}\}$ with $L' = K \times sH \times sW$ and $K \ll T$, and then *plays* by adaptively weighting these entries to produce aggregated cost features that balance contributions across the selected frames. *(6) Iterative Refinement:* following a RAFT-style iterative scheme [32], we alternate GRU-based updates of disparity estimates with PPM-based memory updates, progressively refining $\{d_t\}$ while preserving temporal consistency and computational efficiency .

## 3.2 Memory Pick Process

Naive heuristic strategies, such as random selection or solely keeping the latest frame, are unreliable. Since the former neglects frame reliability and relevance, while the latter suffers from limited temporal context and knowledge drift [36]. To this end, we introduce a Quality Assessment Module (QAM) that explicitly evaluates the quality of memory elements $\{k_m, v_m\}$ in the vanilla buffer for dynamic stereo matching. To activate QAM, we define two complementary scores that quantify each reference frame's contribution to the final accuracy: a confidence score $\mathbf{S}_t^c$ computed over the value embeddings $v_m$ to prioritize reliable evidence, and a redundancy-aware relevance score $\mathbf{S}_t^r$ computed over the key embeddings $k_m$ to suppress repetitive or low-information entries. The full procedure is summarized in Algorithm 1. $\mathbf{S}_t^c$ and $\mathbf{S}_t^r$ are used together to enable the construction of a compact, high-quality memory $\mathcal{M}_t^d$ that preserves the most informative cross-frame cues.

**Confidence Score.** Memory values $v_m$ encode pixel-wise horizontal displacements, which are critical for disparity estimation. These features naturally indicate the reliability of its disparity estimation. To this end, we employ a lightweight confidence network[3] that transforms $v_m \in \mathbb{R}^{T \times sH \times sW \times C}$ into confidence maps $u_t \in \mathbb{R}^{T \times sH \times sW}$, quantifying whether memory values $v_m$ corresponding to

---

[3]The confidence network consists of two convolutional layers followed by a sigmoid activation, which ensures efficient and effective confidence estimation.

---

**Algorithm 1** Pseudo code of Pick-and-Play Memory

---

**Input:** Video frames sequence $\{I_L^t, I_R^t\}$ of video length $T$, GRU $n$-th iterations, $K \ll T$

**Intermediates:** Vanilla Memory: $\mathcal{M} = \left\{ k_m \in \mathbb{R}^{L \times C}, v_m \in \mathbb{R}^{L \times C)} \right\}$, $L = T \times sH \times sW$

   The query: $q_t \in \mathbb{R}^{1 \times sH \times sW}$, $s \in \{1/16, 1/8, 1/4\}$ is the downsampled scale

   Dynamic Memory: $\mathcal{M}_t^d = \left\{ k_m' \in \mathbb{R}^{L' \times C}, v_m' \in \mathbb{R}^{L' \times C)} \right\}$, $L' = K \times sH \times sW$

**Output:** The residual disparity map at $n$-th GRU iteration: $\Delta d_t^n$

---

**1: while** $t \leq T$ **do**

   **Memory Pick Process:**

**2:**   $\mathbf{S}_t = \mathbf{S}_t^c + \mathbf{S}_t^r$   # QAM, evaluate the quality of memory elements $k_m$ and $v_m$

**3:**   $\mathcal{I}_t = \{i \mid \text{rank}\,(\mathbf{S}_t[i]) \leqslant K\}$ # Select top-$K$ reference frames

**4:**   $\mathcal{M}_t^d = \{k_m' = \text{Cat}\,[\{k_i \mid i \in \mathcal{I}_t\}], v_m' = \text{Cat}\,[\{v_i \mid i \in \mathcal{I}_t\}]\}$

   **Memory Play Process:**

**5:**   $\overline{\mathbf{S}}_t[i] = \frac{\mathbf{S}_t[i]}{\sum_i \mathbf{S}_t[i]}, i \in \mathcal{I}_t$   # Balance the contribution of selected memory entries

**6:**   $q_t = q_t + p_t, \quad k_m' = \overline{\mathbf{S}}_t \cdot k_m' + P_{\mathcal{I}_t}$   # Dynamic memory modulation

**7:**   $F_{agg}^t = \text{Read-out}(q_t, \mathcal{M}_t^d)$   # Aggregate high-quality spatio-temporal cost information

**8:**   $\Delta d_t^n = \text{GRU}(F_{agg}^t, F_{cost}^t, F_c^t)$   # Produce the disparity map at the $n$-th iteration

---

accurate disparity outputs. These confidence maps can provide a frame-level reliability measure by estimating the uncertainty of predicted disparity [42, 49]. During training for $N$ iterations, the confidence maps are supervised using an $L_1$ loss function to enforce consistency with their ground-truth counterparts. The ground-truth confidence score $\hat{u}_t$ is computed as follows:

$$\hat{u}_t = \exp\left(-\left|\frac{d_t - \hat{d}_t}{\sigma}\right|\right), \tag{1}$$

where $d_t$ and $\hat{d}_t$ represent the predicted and ground-truth disparities for the $t$-th frame, respectively, and $\sigma$ is a hyper-parameter empirically set to 5. Over $N$ iterations, we compute the confidence loss $L_{conf}$ across all timesteps $u_{t \in (1,T)}$ as follows:

$$\mathcal{L}_{conf} = \sum_{t=1}^{T} \sum_{n=1}^{N} \gamma^{N-n} \|u_t^n - \hat{u}_t^n\|_1, \tag{2}$$

where $n$ denotes the number of iterations and $\gamma$ is a decay factor set as 0.9. To obtain a frame-level confidence score $\mathbf{S}_t^c \in \mathbb{R}^{1 \times T}$, we apply average pooling across the spatial dimensions of the confidence maps $u_t$.

**Redundancy-aware Relevance Score.** Relying solely on the confidence score is insufficient, as adjacent frames often exhibit strong spatio-temporal correlations, which can result in higher confidence scores. This, in turn, introduces feature redundancy and suppresses contributions from more diverse frames, ultimately limiting the diversity and effectiveness of the memory buffer. To mitigate this issue, we propose a redundancy-aware relevance score to evaluate memory keys $k_m$, balancing semantic consistency and memory diversity. First, we compute an inter-frame similarity score $\mathbf{Sim}_t \in \mathbb{R}^{1 \times T}$ between the current query $q_t$ and the memory keys $k_m$, measuring semantic alignment while preserving temporal coherence. For computational efficiency, we employ an attention mechanism combined with spatial downsampling. Specifically, average pooling reduces the spatial resolution of the query and memory keys from $sH \times sW$ to $sH' \times sW'$, followed by L2-normalization

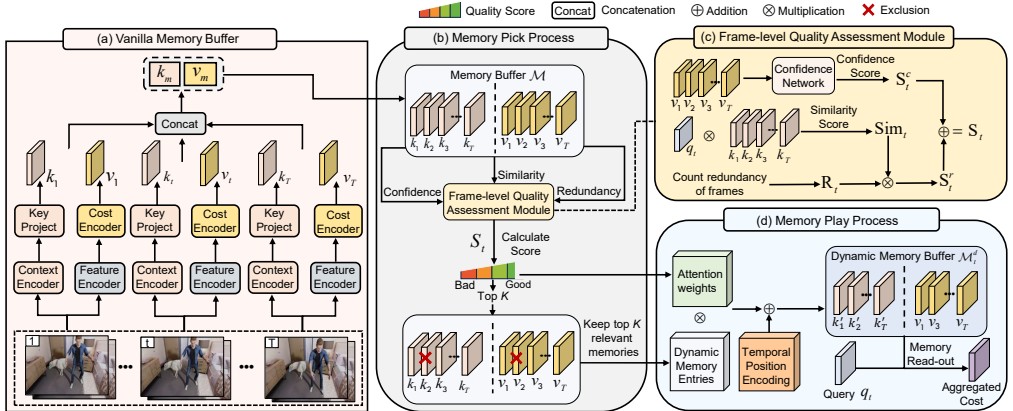

Figure 3: The details of our Pick-and-play Memory Construction Process (PPM).

along the combined feature dimension $f = sH' \times sW' \times C$. The similarity score is computed as:

$$\mathbf{sim}_t = \phi(q_t)\phi(k_m)^T, \text{ where } \phi(x) = \frac{\text{AvgPool}(x)}{||\text{AvgPool}(x)||_2} \qquad (3)$$

where $\phi(k_m) \in \mathbb{R}^{T \times f}$ and AvgPool($\cdot$) denotes the average pooling operation. However, focusing solely on the most similar regions may overlook occluded areas. Since occluded regions in adjacent frames tend to be highly similar, they can be challenging to reference effectively. To mitigate this, we then introduce a redundancy-aware regularizer $\mathbf{R}_t[k] = e^{-\frac{t_k}{T}}$, where $t_k$ denotes the the cumulative number of times the $k$-th frame has been selected for the dynamic memory buffer across previous GRU iterations. This term dynamically downweights overused frames while promoting underutilized yet informative references, ensuring a compact yet diverse memory buffer. The final redundancy-aware relevance score $\mathbf{S}_t^r \in \mathbb{R}^{1 \times T}$ combines redundancy and similarity:

$$\mathbf{S}_t^r = \mathbf{R}_t \cdot \mathbf{sim}_t \qquad (4)$$

By jointly considering relevance and diversity, our approach enhances feature aggregation while minimizing redundancy, leading to more robust and efficient memory-based processing.

**Memory Updating via QAM.** We compute the total quality metric for each memory frame as $\mathbf{S}_t = \mathbf{S}_t^c + \mathbf{S}_t^r$ by integrating confidence and redundancy-aware relevance scores. This integrated scoring enables dynamic memory update by retaining the most informative entries via a top-$K$ selection mechanism, ensuring robust adaptation to varying video scenarios while preventing memory overload. Specifically, for the vanilla memory buffer $\mathcal{M} = \{k_m, v_m\}$ with the corresponding quality scores $\mathbf{S}_t \in \mathbb{R}^{1 \times T}$, we sort the quality scores in descending order and only retain the top-$K$ memory features in the vanilla memory buffer as:

$$\mathcal{I}_t = \{i \mid \text{rank}(\mathbf{S}_t[i]) \leqslant K\} \qquad (5)$$
$$\mathcal{M}_t^d = \{\text{Cat}[\{k_i \mid i \in \mathcal{I}_t\}], \text{Cat}[\{v_i \mid i \in \mathcal{I}_t\}]\}, \qquad (6)$$

{where rank($\cdot$) denotes the ranking position in descending order, with rank = 1 corresponding to the highest score, $\mathcal{I}_t$ is the set of selected frames' indices, and Cat denotes the concatenation. The resulting dynamic memory buffer $\mathcal{M}_t^d$ comprises keys $k_m' = \{k_i\}_{(i \in \mathcal{I}_t)}$, and values $v_m' = \{v_i\}_{(i \in \mathcal{I}_t)}$. By enforcing $K \ll T$, this strategy efficiently handles arbitrary video sequences while providing high-quality spatio-temporal cues for dynamic memory aggregation.

### 3.3 Memory Play Process

After the pick process selects the top-$K$ most relevant memory entries for our dynamic memory buffer $\mathcal{M}_t^d$, we argue that not all selected frames contribute equally to disparity estimation. To further weigh their importance, we introduce a memory play process that dynamically weights the selected memory entries based on learned quality scores. Since dynamic memory construction inherently disrupts temporal ordering, we incorporate temporal position encoding into the framework, ensuring temporal awareness.

**Dynamic Memory Modulation.** Building on this foundation, we propose a unified dynamic memory modulation strategy that jointly optimizes feature reliability and temporal consistency. Specifically, given the estimated quality score $\mathbf{S}_t$, we first obtain the relative significance of the frames:

$$\overline{\mathbf{S}}_t[i] = \frac{\mathbf{S}_t[i]}{\sum_i \mathbf{S}_t[i]}, i \in \mathcal{I}_t \tag{7}$$

Following [16], we initialize positional encodings (PE) to align with the original memory buffer length $T$, formalized as $P_{1:T}$. This initialization ensures temporal coherence in feature representation. Therefore, the 'play' process subsequently operates as follows:

$$q_t = q_t + P_t, \qquad k'_m = \overline{\mathbf{S}}_t \cdot k'_m + P_{\mathcal{I}_t} \tag{8}$$

where $P_t$ denotes the positional encoding at timestep $t$, and $\overline{\mathbf{S}}_t$ represents the aggregated importance weights over the index set $\mathcal{I}_t$. Leveraging the estimated quality scores as reliability indicators, we prioritize more reliable memory entries while maintaining computational efficiency.

**Memory Read-out.** We aggregate cost features through an attention-based memory read-out mechanism from the dynamic memory buffer $\mathcal{M}_t^d$. Specifically, we first compute soft attention weights by measuring the similarity between the query $q_t$ and modulated memory keys $k'_m$. The aggregated cost features $F_{agg}^t$ are then obtained by weighting the memory values $v'_m$ through these attention weights:

$$F_{agg}^t = F_{cost}^t + \alpha \cdot \text{Softmax}\left(1/\sqrt{D_k} \times q_t \times k'^{\mathsf{T}}_m\right) \times v'_m, \tag{9}$$

where $\alpha$ is a learnable scalar initialized from 0. In this way, we employ the attention to gather additional temporal information. With the context, cost, and aggregated cost features, we can now output a residual disparity map through a GRU unit at the $n$-th iteration: $\Delta d_n = \text{GRU}(F_{cost}^t, F_{agg}^t, F_c^t)$. After $N$ iterations of PPM and GRU, we can get the final disparity map.

**Loss Functions.** Our disparity loss functions are inherited from the previous works [24, 22]. Generally, for $N$ iterations, we supervise our network with $L_1$ distance between our a series of residual flows $\{d_1, \ldots, d_T\}$ and the ground-truth $\hat{d}_t$ with exponentially increasing weights:

$$\mathcal{L}_d = \sum_{t=1}^{T} \sum_{n=1}^{N} \gamma^{N-n} \left\| d_t^n - \hat{d}_t \right\|_1, \tag{10}$$

where $\gamma$ and $N$ are set as 0.9 and 10, respectively. Therefore, the total loss function is as follows:

$$\mathcal{L}_{total} = \mathcal{L}_d + \mathcal{L}_{conf}. \tag{11}$$

## 4 Experiments

### 4.1 Datasets

Our work focuses on videos captured with moving cameras, rendering standard image benchmarks like Middlebury [39], ETH3D [40] unsuitable. For training and evaluation, we employ three synthetic and one real-world stereo video dataset, all featuring dynamic scenes: **SceneFlow (SF)** [34] comprising FlyingThings3D, Driving, and Monkaa, with FlyingThings3D featuring moving 3D objects against varied backgrounds. **Dynamic Replica (DR)** [24], a synthetic indoor dataset with non-rigid objects such as people and animals. **Sintel** [6], a synthetic movie dataset available in clean and final passes. **South Kensington (SV)** [23], a real-world stereo dataset without ground truth data, capturing daily scenarios. We use them for generalization evaluation. Following prior work [24, 22], we train on synthetic datasets (SF and DR + SF) and evaluate the performance on **Sintel**, **DR**, and **SV**.

### 4.2 Implementation Details

We implement PPMStereo in PyTorch, training on $8\times$ A100 GPUs (batch size = 2) using $320\times512$ crops from 5-frame sequences, evaluated at full resolution with 20-frame sequences. We use AdamW (lr = 0.0003) with one-cycle scheduling, training for $180k$ iterations ($\approx$ 4.5 days). Data augmentation follows DynamicStereo [24], including random crops and saturation shifts. For efficient memory read-out, we employ FlashAttention [13]. Following prior works [22, 24], we set the number of evaluation

Table 1: Quantitative comparison with SoTA methods. Abbreviations: K - KITTI [35], M - Middlebury [39], ISV–Infinigen SV [23], VK – Virtual KITTI2 [7]. CREStereo utilize 7 datasets for training, including SF [34], Sintel [6], FallingThings [48], InStereo2K [3], Carla [14], AirSim [41], and CREStereo dataset [27]. The best results are in bold, and the second-best are underlined.

| Training data | Method | Sintel Stereo | | | | | | | | Dynamic Replica | | | |
| | | Clean | | | | Final | | | | First 150 frames | | | |
| | | $\delta_{3px}$ | TEPE | $\delta^t_{1px}$ | $\delta^t_{3px}$ | $\delta_{3px}$ | TEPE | $\delta^t_{1px}$ | $\delta^t_{3px}$ | $\delta_{1px}$ | TEPE | $\delta^t_{1px}$ | $\delta^t_{3px}$ |
|---|---|---|---|---|---|---|---|---|---|---|---|---|---|
| SF | CODD [30] | 8.68 | 1.44 | 10.8 | 5.65 | 17.46 | 2.32 | 18.56 | 9.79 | 6.59 | 0.105 | 1.04 | 0.42 |
| | RAFT-Stereo [32] | 6.12 | 0.92 | 9.33 | 4.51 | 10.40 | 2.10 | 13.69 | 7.08 | 5.51 | 0.145 | 2.03 | 0.65 |
| | DynamicStereo [24] | 6.10 | 0.77 | 8.41 | 3.93 | 8.97 | 1.45 | 11.95 | 5.98 | 3.44 | 0.087 | 0.75 | 0.24 |
| | BiDAStereo [22] | 5.94 | 0.73 | 8.29 | 3.79 | 8.78 | 1.26 | 11.65 | 5.53 | 5.17 | 0.103 | 1.11 | 0.40 |
| | **PPMStereo (Ours)** | 5.34 | 0.64 | 7.38 | 3.40 | 7.87 | 1.14 | 10.12 | 4.99 | 2.95 | 0.066 | 0.67 | 0.23 |
| | **PPMStereo_VDA (Ours)** | **4.62** | **0.58** | **6.89** | **3.08** | **7.21** | **1.04** | **9.84** | **4.65** | **2.37** | **0.059** | **0.61** | **0.22** |
| SF + M + K | CODD [30] | 9.11 | 1.33 | 12.16 | 6.23 | 11.90 | 2.01 | 16.16 | 8.64 | 10.03 | 0.152 | 2.16 | 0.77 |
| SF + M | RAFT-Stereo [32] | 5.86 | 0.85 | 8.79 | 4.13 | 8.47 | 1.63 | 12.40 | 6.23 | 3.46 | 0.114 | 1.34 | 0.41 |
| 7 datasets (incl. Sintel) | CREStereo [29] | 4.58 | 0.67 | 6.36 | 3.26 | 8.17 | 1.90 | 12.29 | 6.87 | 1.75 | 0.088 | 0.88 | 0.29 |
| DR + SF | RAFT-Stereo [32] | 5.71 | 0.84 | 9.15 | 4.40 | 9.16 | 2.27 | 13.45 | 7.17 | 1.89 | 0.075 | 0.77 | 0.25 |
| DR + SF | DynamicStereo [24] | 5.77 | 0.76 | 8.46 | 3.93 | 8.68 | 1.42 | 11.93 | 5.92 | 3.32 | 0.075 | 0.68 | 0.23 |
| DR + SF | BiDAStereo [22] | 5.75 | 0.75 | 8.03 | 3.76 | 8.52 | 1.22 | 11.04 | 5.30 | 2.81 | 0.062 | 0.62 | 0.22 |
| DR + SF | **PPMStereo (Ours)** | 5.19 | 0.62 | 7.21 | 3.29 | 7.64 | 1.11 | 9.98 | 4.87 | 2.52 | 0.057 | 0.60 | 0.20 |
| DR + SF | **PPMStereo_VDA (Ours)** | **4.47** | **0.56** | **6.69** | **2.97** | **7.03** | **1.02** | **9.65** | **4.51** | **1.81** | **0.052** | **0.51** | **0.17** |

| Left Frame | DynamicStereo | BidaStereo | **PPMStereo (Ours)** | **PPMStereo_VDA (Ours)** |

Figure 4: Qualitative comparisons on the Sintel final dataset.

iterations $N$ to 20, while setting $N = 10$ during training. Besides, we adopt $n$-pixel error rate ($\delta_{npx}$) for accuracy analysis. Additionally, we use the temporal end-point-error (TEPE) to quantify error variation over time, and $\delta^t_{npx}$ denotes the percentage of pixels with TEPE exceeding $n$ pixels. Lower values on metrics indicate greater temporal consistency and disparity estimation accuracy. Besides, we replace our original feature extractor with Video Depth Anything (ViT-Small) [8]. This PPMStereo_VDA variant leverages pre-trained representations to further boost performance.

## 4.3 Comparison with State-of-the-Art Methods

**Quantitative Results.** As shown in Tab. 1, For the SF version, our PPMStereo achieves state-of-the-art performance, outperforming BiDAStereo [24] by 12.3% & 9.52% and DynamicStereo by 16.8% & 21.3% in TEPE on Sintel clean/final pass. The method also demonstrates strong generalization on Dynamic Replica, surpassing all previous approaches across all metrics. Remarkably, our PPMStereo trained only on synthetic data even largely exceeds the temporal consistency and accuracy of CREStereo [27] on Sintel final pass, despite CREStereo using Sintel data for training. For the SF & DR version, our method achieves superior temporal consistency with a TEPE of 0.057 on Dynamic Replica, significantly outperforming all previous works. Notably, this is achieved with training on only two synthetic datasets, while CREStereo [27] requires seven diverse datasets, demonstrating the efficacy of our long-range temporal modeling. Overall, the results highlight our method's robust performance and generalization ability in both seen and unseen domains. Besides, compared to the previous SoTA method BiDAStereo [22], our method achieves better performance with lower computational costs and memory usage (Please see the appendix for details).

**Qualitative Results.** Our visual comparisons (Fig. 4) using the DR+SF checkpoint show PPMStereo produces sharper disparity predictions than DynamicStereo [24] and BiDAStereo [22], especially in textureless regions (e.g., glass surfaces) where competing methods exhibit blurring artifacts. Besides, following prior work [22, 24], we validate temporal consistency on static scenes by rendering depth point clouds at 15-degree viewpoint increments (Fig. 5). Our method shows significantly smaller high-

Table 2: Ablations of memory buffer module variants trained on DR+SF. 'OOM' denotes CUDA out of memory. 'Baseline' refers to our backbone model without any memory-related modules.

| Experiments | Method | Sintel Final | | Dynamic Replica | |
| | | $\delta_{3px}$ | TEPE | $\delta_{1px}$ | TEPE |
|---|---|---|---|---|---|
| | Baseline | 8.65 | 1.37 | 3.10 | 0.074 |
| Memory Buffer | Full | OOM | | OOM | |
| | MemFlow [15] | 8.45 | 1.28 | 3.11 | 0.070 |
| | Latest | 8.11 | 1.19 | 2.89 | 0.062 |
| | Random | 8.42 | 1.26 | 2.99 | 0.064 |
| | XMem [9] | 8.04 | 1.18 | 2.84 | 0.061 |
| | RMem [66] | 7.93 | 1.16 | 2.77 | 0.061 |
| | **Ours** | **7.64** | **1.11** | **2.52** | **0.057** |
| Memory Length | $K = 1$ | 7.95 | 1.18 | 2.70 | 0.062 |
| | $K = 3$ | 7.80 | 1.13 | 2.58 | 0.057 |
| | $K = 5$ | 7.64 | 1.11 | 2.52 | 0.057 |
| | $K = 7$ | **7.62** | **1.10** | **2.50** | **0.057** |

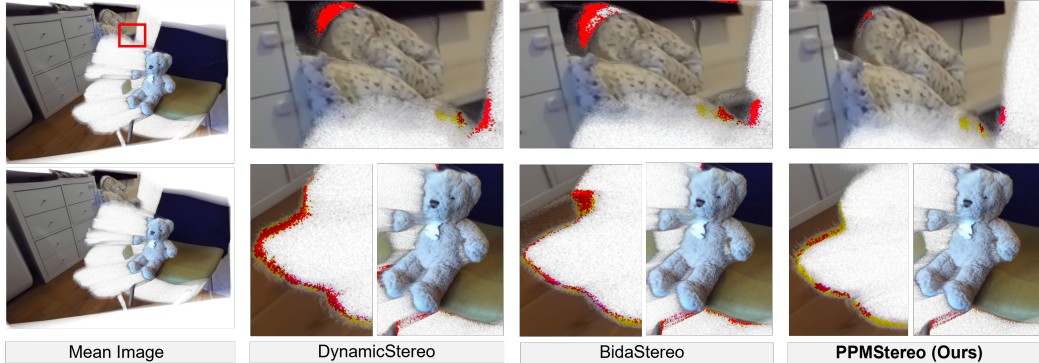

| Mean Image | DynamicStereo | BidaStereo | **PPMStereo (Ours)** |

Figure 5: Temporal consistency comparison on 50-frame reconstructed stereo video (all trained on DR + SF). Our method achieves lower variance, demonstrating superior consistency.

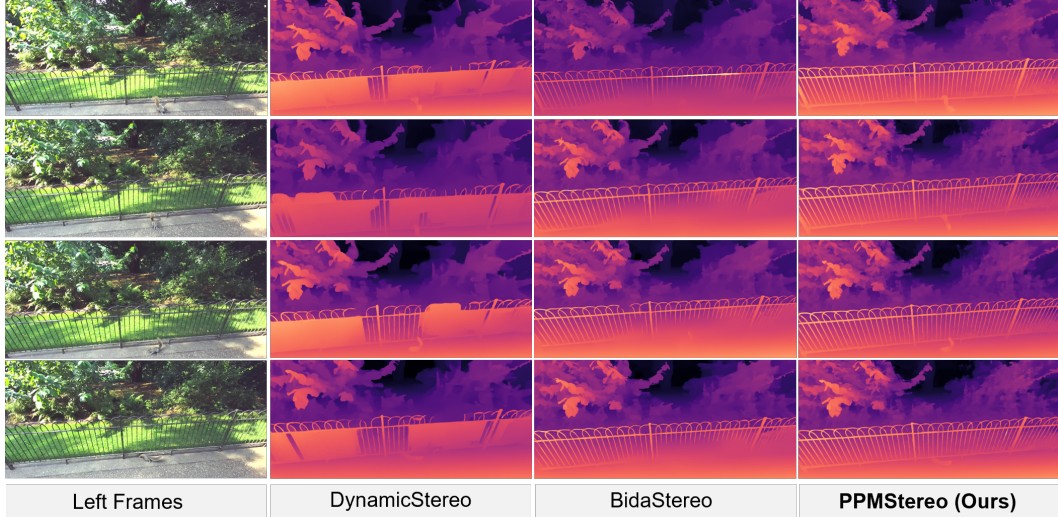

| Left Frames | DynamicStereo | BidaStereo | **PPMStereo (Ours)** |

Figure 6: Qualitative generalization comparison on a dynamic outdoor scenario from the SV dataset.

variance regions ($> 40\ px^2$, marked red), confirming superior stability. Furthermore, on the real-world outdoor scenes from the South Kensington dataset [23] (Fig. 6), PPMStereo accurately recovers thin structures such as the fences while maintaining temporal consistency, demonstrating robust generalization to unseen domains. More visualizations are provided in the appendix.

## 4.4 Ablation Studies

Due to the huge training cost of PPMStereo_VPA, we conduct ablation studies exclusively on PPMStereo below. Besides, all ablated models below are trained on DR + SF.

**Memory buffer construction.** We train and evaluate 5 different memory buffer variants, namely, keeping frames from (1) full frames (20 frames), (2) MemFlow (1 frame) [15], (3) the latest frames (5 frames), (4) random (5 frames), (5) XMem [9] (distilling all outdated memory features into long-term memory based on attention scores), (6) RMem [66] (5 frames), and (7) ours (5 frames).

Specifically, we replace the memory buffer variants and keep the remaining modules unchanged during training and inference. Table 2 shows three key insights: First, while reference frames improve performance, naive accumulation shows diminishing returns, indicating memory capacity alone is insufficient. Second, frame selection quality critically affects results. The random selection policy underperforms even single-neighbor memory (MemFlow) [15] on Sintel final pass, highlighting selection importance. However, on the DR dataset with minimal inter-frame changes, the random policy performs comparably to advanced variants. Lastly, direct long-term memory integration

Table 3: Ablation Study of PPM on Sintel and Dynamic Replica. All models are trained on DR+SF. Note that we directly perform the read-out operation for the ablated model without the 'play' process.

| ID | Pick-and-Play Memory | | Sintel Final | | | | Dynamic Replica | | | |
|---|---|---|---|---|---|---|---|---|---|---|
| | Pick | Play | $\delta_{3px}$ | TEPE | $\delta^t_{1px}$ | $\delta^t_{3px}$ | $\delta_{1px}$ | TEPE | $\delta^t_{1px}$ | $\delta^t_{3px}$ |
| 1 | Baseline | | 8.65 | 1.37 | 11.72 | 5.91 | 3.10 | 0.074 | 0.72 | 0.23 |
| 2 | ✓ | | 7.81 | 1.14 | 10.24 | 5.07 | 2.65 | 0.060 | 0.64 | 0.21 |
| 3 | | ✓ | 7.97 | 1.17 | 10.36 | 5.20 | 2.80 | 0.062 | 0.68 | 0.21 |
| 4 | ✓ | ✓ | **7.64** | **1.11** | **9.98** | **4.87** | **2.52** | **0.057** | **0.60** | **0.20** |

Table 4: Ablation study on the 'pick' process. C, Sim, and R denote confidence score, similarity score, and redundancy factor, respectively.

| ID | QAM | | | Sintel Final | | | Dynamic Replica | | |
|---|---|---|---|---|---|---|---|---|---|
| | C | Sim | R | $\delta_{3px}$ | TEPE | $\delta^t_{3px}$ | $\delta_{1px}$ | TEPE | $\delta^t_{1px}$ |
| 1 | Baseline | | | 7.97 | 1.17 | 5.20 | 2.80 | 0.062 | 0.68 |
| 2 | ✓ | | | 7.81 | 1.14 | 5.06 | 2.63 | 0.058 | 0.65 |
| 3 | ✓ | ✓ | | 7.74 | 1.12 | 4.95 | 2.57 | 0.057 | 0.62 |
| 4 | ✓ | ✓ | ✓ | **7.64** | **1.11** | **4.87** | **2.52** | **0.057** | **0.60** |

Table 5: Ablation study on the 'play' process. Weights and PE denote the weighting operation and the temporal position encoding, respectively.

| ID | Play Process | | Sintel Final | | | Dynamic Replica | | |
|---|---|---|---|---|---|---|---|---|
| | Weights | PE | $\delta_{3px}$ | TEPE | $\delta^t_{3px}$ | $\delta_{1px}$ | TEPE | $\delta^t_{1px}$ |
| 1 | Baseline | | 7.81 | 1.14 | 5.07 | 2.65 | 0.060 | 0.64 |
| 2 | ✓ | | 7.67 | 1.12 | 5.00 | 2.54 | 0.060 | 0.62 |
| 3 | | ✓ | 7.77 | 1.11 | 4.93 | 2.63 | 0.058 | 0.61 |
| 4 | ✓ | ✓ | **7.64** | **1.11** | **4.87** | **2.52** | **0.057** | **0.60** |

(XMem) shows limited impact, suggesting that simply using all frames may be less effective than the RMem variant. In contrast, our PPM mechanism overcomes these limitations by dynamically identifying and modulating valuable reference frames, achieving significant TEPE improvements on these two datasets (+19.0% TEPE on Sintel and +22.9% TEPE on DR) over the baseline.

**Memory length.** Table 2 shows the impact of memory length on PPMStereo. Performance improves initially (e.g., +14.8% $\delta^t_{1px}$ on Sintel for $K \leq 5$) when trained and evaluated at this memory length, but performance saturates beyond $K = 5$ due to feature redundancy. To balance computational efficiency and model accuracy, we select $K = 5$ as the optimal memory length for our final model.

**Contribution of each component.** Table 3 shows the proposed PPM module outperforms window-based aggregation through two key processes: (1) The pick process dynamically selects high-quality memory elements from non-adjacent frames, overcoming fixed-window limitations and improving occlusion handling; (2) The play process adaptively weights features by semantic relevance, reducing noise propagation (ID = 3 shows +0.2 on Sintel and +0.017 TEPE improvements on DR compared to the baseline). By combining them, they provide complementary benefits. The pick ensures feature diversity while play suppresses outliers, yielding superior performance in dynamic stereo matching.

**QAM.** Our QAM module dynamically assesses frame reliability in the memory buffer using a scoring mechanism. We refresh the memory buffer by balancing: (1) cost feature quality ($v_m$) and (2) redundancy-aware semantic relevance ($k_m$) (Sec. 3.2). Table 4 shows that our quality score improves both depth accuracy and temporal consistency. Fig. 7 further confirms the confidence map's strong correlation with the error map, validating it as a reliable quality indicator for $v_m$.

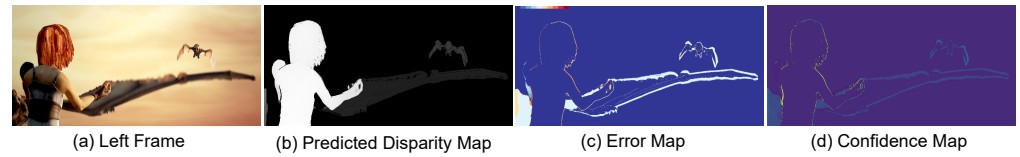

(a) Left Frame  (b) Predicted Disparity Map  (c) Error Map  (d) Confidence Map

Figure 7: Visualization of error map and confidence map. Brighter regions denote higher uncertainty.

**Memory modulation.** Our proposed memory modulation mechanism (Sec. 3.3) further enhances spatio-temporal modeling, achieving a performance gain with +0.17 $\delta_{3px}$ and +0.13 $\delta_{1px}$ improvements on the Sintel Final and DR, respectively, as seen in Table 5. The adaptive weighting mechanism dynamically prioritizes the most important spatio-temporal features, highlighting accuracy improvements. Meanwhile, learned positional embeddings endow the model with temporal awareness, improving the overall temporal consistency. Experiments show that these components work together to strengthen the model's ability to capture long-range dependencies and distinguish key spatio-temporal patterns.

# 5 Conclusion

In this paper, we introduce PPMStereo, the first framework, to our knowledge, to leverage high-quality memory for dynamic stereo matching. By selectively updating and modulating the most valuable memory entries, our proposed pick-and-play memory construction mechanism enables the integration of cost information across long-range spatio-temporal connections, ensuring temporally consistent stereo matching. Extensive experiments demonstrate the effectiveness of our approach across diverse datasets, highlighting its generic applicability.

# Acknowledgment

This work was partly supported by the Shenzhen Science and Technology Program under Grant RCBS20231211090736065, GuangDong Basic and Applied Basic Research Foundation under Grant 2023A1515110074. This work was also supported by the InnoHK Initiative of the Government of the Hong Kong SAR and the Laboratory for Artificial Intelligence (AI)-Powered Financial Technologies, with additional support from the Hong Kong Research Grants Council (RGC) grant C1042-23GF and the Hong Kong Innovation and Technology Fund (ITF) grant MHP/061/23.

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
