# OpenReview forum: "PPMStereo: Pick-and-Play Memory Construction for Consistent Dynamic Stereo Matching"
_NeurIPS.cc/2025/Conference — NeurIPS 2025 poster_

### Official Review · Reviewer_Y5FR · 2025-06-19

**Clarity:** 3
**Significance:** 3
**Originality:** 3
**Rating:** 5
**Confidence:** 4

**Summary:**

This paper proposes a novel dynamic stereo architecture that uses a memory bank approach with a custom Quality Assessment Module (QAM) to achieve state-of-the-art performance.
In particular, the authors propose an iterative-based (i.e., based on RAFT-Stereo architecture) dynamic stereo model that uses features not only from the cost volume and context image, but also from a Dynamic Memory Buffer.
The latter is used to store temporal consistency information, built using the proposed Memory Pick'n'Play Processes, that ensure a selection of the most relevant temporal information.
The memory bank approach is used not only to store features from different temporal frames but also to reduce memory consumption via a frame selection strategy, based on the outcome of the QAM module.
Exhaustive experiments prove the effectiveness of the architecture design and the performance of the model w.r.t. the previous state of the art.

**Questions:**

Before asking questions, I would like to introduce my rationale behind my initial rating: the paper proposes an incremental work, extending MemFlow for Dynamic Stereo Matching and introducing a revisited memory bank dubbed as Pick'n'Play Memory. The exhaustive experiments show SOTA performance, thus confirming a valid contribution for NIPS 2025.

- Inside equation 8, there is $k'_m$ on both parts of the right equation. Is it right?
- How $T$ -- a.k.a. number of reference frames -- hyper-parameter is chosen?
- Please refer to Strengths and Weaknesses for further suggestions.

**Ethical Concerns:**

["NO or VERY MINOR ethics concerns only"]

**Final Justification:**

After reading the rebuttal and other reviews, I would like to suggest the acceptance of the paper to NIPS 2025.

The reply made by the authors resolves all my initial concerns.

**Limitations:**

yes

**Quality:**

3

**Strengths And Weaknesses:**

Strengths

- **Exhaustive experiments showing SOTA performance**: The authors validate the potential of their proposal using different standard benchmarks such as Sintel, Dynamic Replica (Tab. 1). Furthermore, they qualitatively investigate in Fig. 6 the performance using a real-world dataset -- i.e., South Kensington. This largely proves the effectiveness of the proposed Pick'n'Play strategy.
- **Novel Dynamic Stereo Architecture**: The authors revisited the MemFlow architecture for Dynamic Stereo Matching. Additionally, they changed the memory bank module with the improved Pick'n'Play proposal. The benefits of the latter solution w.r.t. MemFlow memory bank are shown in Table 2.
- **Clear logic structure of the paper**: I personally liked the clear and easy-to-follow structure of the paper. The authors shortly introduce the concept of iterative stereo networks -- assuming a prior knowledge from the reader -- while putting a lot of effort into the Pick'n'Play processes, the core idea of the paper. Even so, chapter 3 was a little hard to digest at first: I suggest that the authors make more references to Fig. 2 and Fig. 3 inside the chapter.

Weaknesses

- **Related works regarding Deep Stereo Matching can be enlarged**: In my opinion, the related works regarding Deep Stereo Matching are quite poor, since the authors mainly report the GRU-based architectures. I suggest at least including a survey paper [1] to include additional literature regarding deep stereo matching.
- **Missing competitors with VDA features**: The benefits of the VDA encoder are quite impressive. It is true that authors showed SOTA performance even without the VDA encoder; however, I would have liked to see a fairer comparison showing at least one competitor with the VDA encoder.
- **The paper lacks a runtime memory consumption**: I suggest that the authors enrich Fig. 7 of the appendix by adding a measure of memory consumption during evaluation.


[1] Tosi, Fabio, Luca Bartolomei, and Matteo Poggi. "A survey on deep stereo matching in the twenties." International Journal of Computer Vision (2025): 1-32.

---

> ### Author Rebuttal · Authors · 2025-07-31
>
> Thank you for reviewing our paper. We appreciate your valuable feedback and will try to address your concerns below:
>
> **S3:** Clear logic structure of the paper: The authors shortly introduce the concept of iterative stereo networks -- assuming a prior knowledge from the reader -- while putting a lot of effort into the Pick'n'Play processes, the core idea of the paper. Even so, chapter 3 was a little hard to digest at first: I suggest that the authors make more references to Fig. 2 and Fig. 3 inside the chapter.
>
> **R3:** Thank you for recognizing the clear logical structure of the paper. For your suggestion that the authors should make more references to Fig. 2 and Fig. 3 inside the chapter, we will organize Section 3.1 into the following coherent workflow:
>
> 1. Feature Extraction
>
> 2. Cost‑Volume Construction
>
> 3. Context Encoding
>
> 4. Memory Buffer Initialization
>
> 5. Pick‑and‑Play Memory Module
>
> 6. Iterative Refinement
>
> In Section 3.2, we will restructure the description of the "pick process" using Figures 3 (b) and 3 (c) to enhance clarity.
>
> In Section 3.3, we will reintroduce the  "play process"  to make it align with the workflow in Figure 3 (d).
> This revised structure will guide the reader from the overall architecture to progressively more detailed components, helping to clarify our design details and contributions.
>
> **W1:** Related works regarding Deep Stereo Matching can be enlarged: In my opinion, the related works regarding Deep Stereo Matching are quite poor, since the authors mainly report the GRU-based architectures. I suggest at least including a survey paper [1] to include additional literature regarding deep stereo matching.
> [1] Tosi, Fabio, Luca Bartolomei, and Matteo Poggi. "A survey on deep stereo matching in the twenties." International Journal of Computer Vision (2025): 1-32.
>
> **A1:** Thank you for your comments regarding the related works on deep stereo matching. We will revise the related work section accordingly and include the following additional reference as follows:
>
> "**Deep Stereo Matching.** The field of deep stereo matching, as comprehensively surveyed by Tosi et al. (2025) [1], has undergone rapid evolution, particularly in network design focused on cost volume construction and aggregation. These methods are generally categorized into regression-based approaches (e.g., Guo et al. (2023) [2]; Guo et al. (2025) [3]) and iterative refinement paradigms. Further developments continue along these directions. and so on."
>
> If you believe the current references are still insufficient, we would sincerely appreciate any additional suggestions.
>
> **W2:** Missing competitors with VDA features: The benefits of the VDA encoder are quite impressive. It is true that authors showed SOTA performance even without the VDA encoder; however, I would have liked to see a fairer comparison showing at least one competitor with the VDA encoder.
>
> **A2:** Thank you for highlighting this important issue. To address your concern regarding the impact of the VDA encoder, we conduct additional experiments by integrating the VDA encoder into previous SoTA models to enable a fairer comparison.
> As shown in the table below, incorporating the VDA encoder into SoTA models consistently improves performance across all benchmarks and metrics:
> |Methods| Sintel Clean (EPE) | Sintel Clean (TEPE)| Sintel Clean (Tδ3px) | Sintel Final (EPE) | Sintel  Final (TEPE) | Sintel  Final (Tδ3px) |
> |:-----------------:| :---------------:|:--------------:|:----------------:| :-----------------------:|:----------------:| :-----------------------:|
> |DynamicStereo|1.34|0.76|3.93 |3.12|1.42|5.92|
> |DynamicStereo_VDA|1.02|0.65 |3.47|2.55|1.20|5.38|
> |BidaStereo|1.38|0.75|3.76 |2.04|1.22|5.30|
> |BidaStereo_VDA|1.04|0.65 | 3.41 |1.67|1.14|4.94|
> |PPMStereo|0.96|0.62|3.29|1.49|1.11|4.87|
> |PPMStereo_VDA|**0.81**|**0.56**|**2.97**|**1.24**|**1.02**|**4.51**|
>
> Remarkably, the proposed method, PPMStereo, still achieves SoTA results even without the VDA encoder, outperforming these VDA-enhanced baselines. When combined with the VDA encoder, PPMStereo further improves and achieves impressive performance. We attribute this performance to the strong priors provided by the Visual Foundation Model.
> These results highlight the strength and flexibility of our Pick-and-Play Memory mechanism, which effectively captures long-range temporal consistency for consistent dynamic stereo matching.
>
> **W3:** The paper lacks a runtime memory consumption: I suggest that the authors enrich Fig. 7 of the appendix by adding a measure of memory consumption during evaluation.
>
> **A3:** We thank the reviewer for the helpful suggestion. To measure runtime memory during evaluation, we conduct evaluation on a 20‑frame sequence from the Sintel Stereo dataset (resolution 1024×436) on an NVIDIA A100, varying the number of iterations *N* up to 20. The table reports peak GPU memory (in GB) and TEPE on the Clean and Final subsets:
>
> |Methods|Iterations|GPU Memory (GB)|Inference Time (s)|Sintel Clean (TEPE)|Sintel Final (TEPE)|
> | :-----------------: | :---------------: | :--------------: | :--------------: |:--------------: |:--------------: |
> |DynamicStereo |*N=20*|**16.6**|3.4|0.76|1.42|
> |BidaStereo |*N=20*|19.8|6.7|0.75|1.22|
> |Ours |*N=5*|20.7|**3.0**|0.72|1.22|
> |Ours |*N=10*|21.9|6.0|**0.63**|**1.12**|
> |Ours |*N=15*|23.4|8.9|**0.62**|**1.11**|
> |Ours |*N=20*|24.4|11.8|**0.62**|**1.11**|
>
> As observed, our approach achieves state-of-the-art accuracy (lowest TEPE) on both Sintel Clean and Final benchmarks, with a gradual increase in GPU memory usage and inference time as *N* increases. Notably, for smaller  *N* (e.g., *N*=5), our method matches the accuracy of prior art with only a moderate increase in memory, while larger *N*  provides further accuracy gains at the cost of higher resource usage.
> To better illustrate this trade-off, we will enrich Fig.7 to include a comparative analysis of runtime memory usage.
>
> **Q1:** Inside equation 8, there is $k'_m$  on both parts of the right equation. Is it right?
>
> **R1:** Thank you for pointing out the typo. We sincerely apologize for the oversight.
> We will revise the symbols on the left-hand side of Eq.8 ($\hat{k}_{m} = \overline{\mathrm{S}}_t \cdot k'_m $ +  temporal_postion_encoding) to eliminate any ambiguity and will also update Eq.9 accordingly.
>
>
>
> **Q2:** How  $T$ -- a.k.a. number of reference frames -- hyper-parameter is chosen?
>
> **R2:** Thank you for your concern regarding the selection of reference frame numbers. To ensure a fair comparison, we follow the configuration adopted in prior works (e.g., DynamicStereo, BiDAStereo), using 5 reference frames during training and 20 during inference.
> Additionally, we conduct further evaluations with varying numbers of reference frames to evaluate their impact. The results are summarized in the table below:
> |Number of Reference Frames|Sintel Clean(TEPE)|Sintel Clean(Tδ1px)|Sintel Clean(Tδ3px)|Sintel Final (TEPE)|Sintel Final(Tδ1px)|Sintel Final(Tδ3px)
> | :----------------: | :-----------------------: | :----------------: | :-----------------------: |:-----------------------: |:-----------------------: |:-----------------------: |
> |$T$ =20|0.62|7.21|3.29|1.11|9.98|4.87|
> |$T$ =30|0.61|7.17|3.29|1.09|9.92|4.81|
> |$T$ =40|0.61|7.17|3.28|1.09|9.93|4.82|
> |$T$ =50|0.61|7.18|3.28|1.09|9.92|4.82|
>
> The table shows that performance improves as the number of frames increases, but eventually reaches a saturation point.

---

> > ### Comment · Reviewer_Y5FR · 2025-08-01
> >
> > Dear Authors,
> >
> > Thanks for the detailed response.
> > I'm satisfied with all the answers.
> >
> > I also acknowledge that I have read the other review and responses.
> >
> > If authors still have some space for the related works on deep stereo matching, I would like to suggest adding this recent trend of Mono+Stereo fusion architectures:
> >
> > - Wen, Bowen, et al. "Foundationstereo: Zero-shot stereo matching." Proceedings of the Computer Vision and Pattern Recognition Conference. 2025.
> > - Bartolomei, Luca, et al. "Stereo anywhere: Robust zero-shot deep stereo matching even where either stereo or mono fail." Proceedings of the Computer Vision and Pattern Recognition Conference. 2025.
> >
> > Best regards,
> >
> > Reviewer Y5FR

---

> ### Author Response · Authors · 2025-08-01
>
> We sincerely thank Reviewer Y5FR for the positive feedback and for acknowledging the other reviews and responses. We also appreciate the valuable suggestion regarding recent advances in deep stereo matching.  Following your recommendation, we will revise the Related Work section to include the following references and expanded discussion:
>
> "**Deep Stereo Matching.** The field of deep stereo matching, as comprehensively surveyed by Tosi et al. (2025) [1], has undergone rapid evolution, particularly in network designs focused on cost volume construction and aggregation. These methods are generally categorized into regression-based approaches (e.g., Guo et al. (2023) [2]; Guo et al. (2025) [3]) and iterative refinement paradigms (Wen et al. [4], Bartolomei et al. [5]). And so on."
>
> We will further elaborate on these developments in the revised manuscript. We believe this addition will further strengthen the coverage of recent trends in deep stereo matching.
>
> [1] Tosi, Fabio, Luca Bartolomei, and Matteo Poggi. "A survey on deep stereo matching in the twenties." IJCV (2025): 1-32.
>
> [2] "Openstereo: A comprehensive benchmark for stereo matching and strong baseline." arXiv:2312.00343 (2025).
>
> [3] "Lightstereo: Channel boost is all you need for efficient 2D cost aggregation." ICRA (2025).
>
> [4] Wen, Bowen, et al. "Foundationstereo: Zero-shot stereo matching." CVPR (2025).
>
> [5] Bartolomei, Luca, et al. "Stereo anywhere: Robust zero-shot deep stereo matching even where either stereo or mono fail." CVPR (2025).

---

### Official Review · Reviewer_6N4q · 2025-06-30

**Clarity:** 3
**Significance:** 3
**Originality:** 3
**Rating:** 4
**Confidence:** 4

**Summary:**

The paper presents a novel framework called PPMStereo for dynamic stereo matching with temporally consistent disparity estimation in video sequences. The authors address the challenge of capturing long-range temporal relationships efficiently by introducing a Pick-and-Play Memory (PPM) module. The system operates through two primary stages:

Pick: Identifying relevant frames based on a Quality Assessment Module (QAM).
Play: Adapting the aggregation of features from selected frames to achieve accurate and temporally consistent disparity maps.

**Questions:**

Please refer to weakness

**Ethical Concerns:**

["NO or VERY MINOR ethics concerns only"]

**Final Justification:**

Thanks for the response. I am satisfied with the authors' response, so I maintain my opinion of accepting the paper.

**Limitations:**

yes

**Quality:**

3

**Strengths And Weaknesses:**

Strengths:
1.Innovative Memory Mechanism: The introduction of a memory buffer for spatio-temporal consistency in dynamic stereo matching is novel. The "Pick-and-Play" strategy improves computational efficiency by dynamically selecting and aggregating relevant frames.
2.Strong Experimental Results: The method achieves state-of-the-art performance on multiple dynamic stereo benchmarks, including Sintel, Dynamic Replica, and Sintel Final Pass. Notably, it outperforms previous methods such as BiDAStereo and DynamicStereo in terms of temporal consistency and accuracy with reduced computational cost.The ablation studies are comprehensive, demonstrating the impact of different components of the framework.
3.Generalization across Datasets: The method demonstrates strong generalization, with superior results on both synthetic and real-world datasets.

Weaknesses:
1.In Table 1, the Memory Length decreases gradually from 1 to 7. The reviewers would like to know the results when k is larger, such as k=9 and k=11.
2.Computational Efficiency: A more detailed discussion of computational costs, including GPU memory usage and processing time per frame or video sequence, would help clarify the trade-offs in terms of efficiency versus accuracy.
3.In the "Deep Stereo Matching" section of the related work, there is a lack of citations to some latest papers:
1） "Openstereo: A comprehensive benchmark for stereo matching and strong baseline." arXiv preprint arXiv:2312.00343 (2025).
2）"Lightstereo: Channel boost is all you need for efficient 2d cost aggregation." (ICRA 2025).

---

> ### Author Rebuttal · Authors · 2025-07-31
>
> Thank you for reviewing our paper. We appreciate your valuable feedback and will try to address your concerns below:
>
> **W1:** In Table 1, the Memory Length decreases gradually from 1 to 7. The reviewers would like to know the results when k is larger, such as k=9 and k=11.
>
> **A1:** We thank you for the helpful suggestion.  In response, we conduct our experiments on larger memory lengths ($K$ = 9, 11). The result is shown in the table below:
>
> | Memory Length (*K*) | Sintel  (δ3px) | Sintel  (TEPE) | Dynamic Replica (δ1px) | Dynamic Replica (TEPE) |
> | :-----------------: | :---------------: | :--------------: | :----------------: | :-----------------------: |
> |Baseline|8.65|1.37|3.10|0.074|
> |5| 7.64 |1.11|2.52|0.057|
> |7| 7.62 |1.10|2.50|0.057|
> |9| **7.61**|**1.10**|**2.49**|**0.057**|
> |11|7.63|1.12|2.52|0.057|
>
> We observe that increasing the memory length beyond 5 offers only marginal gains or even slight degradations when $K=11$. While intuitively, a longer memory bank might appear beneficial by providing a richer temporal context, it can inadvertently introduce redundant or less relevant information. This surplus of data may obscure the salient cues necessary for effective cost aggregation, ultimately impairing the model’s discriminative capability.  This observation is consistent with findings from RMem [1], which reported that overly long temporal contexts can result in diminishing returns.
> These results highlight the importance of balancing temporal diversity with information relevance when designing memory systems. Accordingly, we adopt a memory length of $K = 5$ in our implementation, as it offers a good trade-off between accuracy and computational efficiency.
>
> [1] Zhou J, Pang Z, Wang Y X. Rmem: Restricted memory banks improve video object segmentation[C]//Proceedings of the IEEE Conference on Computer Vision and Pattern Recognition. 2024: 18602-18611.
>
> **W2:** Computational Efficiency: A more detailed discussion of computational costs, including GPU memory usage and processing time per frame or video sequence, would help clarify the trade-offs in terms of efficiency versus accuracy.
>
> **A2:** We appreciate your feedback regarding the computational efficiency. To more clearly illustrate the trade-offs between efficiency and accuracy, we measure GPU memory usage and processing time per video sequence during inference. Specifically, we conduct evaluations on a 20-frame video sequence from the Sintel Final Stereo dataset (resolution: 1024×436) using a single NVIDIA A100 GPU, varying the number of iterations *N*. The results are summarized in the table below:
>
> |Methods|Iterations|GPU Memory (GB)|Inference Time (s)|Sintel Final (TEPE)|Sintel Final (Tδ1px)|Sintel Final (Tδ3px)|
> | :-----------------: | :---------------: | :--------------: | :--------------: | :--------------: | :--------------: |  :--------------: |
> |DynamicStereo |*N*=20|**16.6**|3.4|1.42|11.93 | 5.92 |
> |BidaStereo |*N*=20|19.8|6.7|1.22|11.04| 5.30 |
> |Ours |*N*=5|20.7|**3.0**|1.22|11.84|5.36|
> |Ours |*N*=10|21.9|6.0|1.12|10.10|4.94|
> |Ours |*N*=15|23.4|8.9|**1.11**|9.99| 4.89|
> |Ours |*N*=20|24.4|11.8|**1.11**|**9.98**|**4.87**|
>
> While previous approaches require a large number of iterations (*N*=20) to achieve optimal accuracy, our method, as shown above, attains comparable accuracy while requiring fewer inference iterations (e.g., at *N*=5). Under the same inference iterations, our method largely outperforms the SoTA method BidaStereo across all three evaluation metrics. These results indicate that our approach strikes an effective balance between efficiency and accuracy. We will report these results in the revised manuscript to more clearly clarify our contributions.
> Thanks to the PPM mechanism, our PPMStereo requires fewer iterations to converge. Notably, even with just 5 iterations (*N*=5), it achieves better temporal consistency than DynamicStereo, clearly demonstrating the effectiveness of the proposed method.
>
> **W3:** In the "Deep Stereo Matching" section of the related work, there is a lack of citations to some latest papers: 1） "Openstereo: A comprehensive benchmark for stereo matching and strong baseline." arXiv preprint arXiv:2312.00343 (2025). 2）"Lightstereo: Channel boost is all you need for efficient 2d cost aggregation." (ICRA 2025).
>
> **A3:** Thank you for the valuable suggestion. In the revised manuscript, we will add these citations to the Related Work section and include a corresponding discussion.  If you believe the current references are still insufficient, we would sincerely appreciate any additional suggestions.

---

> > ### Comment · Reviewer_6N4q · 2025-08-02
> >
> > Thanks for the response.  I am satisfied with the authors' response, so I maintain my opinion of accepting the paper. The reviewer suggests that the authors include StereoAnything in the related work section in the final version:
> > Guo X, Zhang C, Zhang Y, et al. Stereo anything: Unifying stereo matching with large-scale mixed data[J]. arXiv preprint arXiv:2411.14053, 2024.

---

> ### Author Response · Authors · 2025-08-02
>
> We sincerely appreciate your encouraging feedback. Thank you also for recommending StereoAnything (Guo et al., 2024). As suggested, we will include these citations in the Related Work section. Specifically, following your recommendation, we will revise the Related Work section to include the following references and expanded discussion:
>
> "**Deep Stereo Matching.** The field of deep stereo matching, as comprehensively surveyed by Tosi et al. (2025) [1], has undergone rapid evolution, particularly in network designs focused on cost volume construction and aggregation. These methods are generally categorized into regression-based approaches (e.g., Guo et al. (2023) [2]; Guo et al. (2025) [3]) and iterative refinement paradigms (Wen et al. [4], Bartolomei et al. [5], Guo et al. [6]). And so on."
>
> We will further elaborate on these developments in the revised manuscript. We believe this addition will further strengthen the coverage of recent trends in deep stereo matching.
>
> [1] Tosi, Fabio, Luca Bartolomei, and Matteo Poggi. "A survey on deep stereo matching in the twenties." IJCV (2025): 1-32.
>
> [2] "Openstereo: A comprehensive benchmark for stereo matching and strong baseline." arXiv:2312.00343 (2025).
>
> [3] "Lightstereo: Channel boost is all you need for efficient 2D cost aggregation." ICRA (2025).
>
> [4] Wen, Bowen, et al. "Foundationstereo: Zero-shot stereo matching." CVPR (2025).
>
> [5] Bartolomei, Luca, et al. "Stereo anywhere: Robust zero-shot deep stereo matching even where either stereo or mono fail." CVPR (2025).
>
> [6]  Stereo anything: Unifying stereo matching with large-scale mixed data[J]. arXiv preprint arXiv:2411.14053, 2024.
>
> Should you have any additional references or insights that could further strengthen the manuscript, we would be grateful to receive them.

---

### Official Review · Reviewer_HYhV · 2025-07-03

**Clarity:** 2
**Significance:** 2
**Originality:** 3
**Rating:** 5
**Confidence:** 3

**Summary:**

This paper proposes a method to improve temporal consistency by capturing long-range dependencies across video frames while maintaining computational efficiency through a novel memory buffer mechanism. The approach first selects the most relevant frames using confidence scores and redundancy-aware relevance metrics, then adaptively weights and aggregates these frames for spatio-temporal feature integration. Experimental results show the proposed method outperforms existing approaches across three benchmark datasets in both accuracy and temporal consistency.

**Questions:**

1）The paper's organization is somewhat disorganized.
2）The excessive use of mathematical notation hinders readability.
3）The paper reads more like a technical report, primarily focusing on implementation details while providing limited explanation of the underlying motivations and theoretical principles.

**Ethical Concerns:**

["NO or VERY MINOR ethics concerns only"]

**Final Justification:**

My main concerns are well addressed. Therefore, I recommend accepting this paper.

**Limitations:**

Does the keyframe extraction and compression process affect inference speed?
How many frames can the performance improvement from long-term temporal information be sustained? The paper only tests up to 20 frames?

**Quality:**

2

**Strengths And Weaknesses:**

Strengths

1）The proposed method demonstrates significant performance improvements.
2）The experiments are comprehensive, providing detailed validation of the method's effectiveness.

Weaknesses
1）The paper's organization is somewhat disorganized. The core idea—selecting and aggregating key frames to balance long-term temporal modeling and computational efficiency could be presented more clearly. A top-down structure (general to specific) would improve readability, rather than introducing excessive details and equations in Section 3.1.
2）The excessive use of mathematical notation hinders readability. Additionally, some equations (e.g., (1) and (2)) are overly simplistic and could be presented inline rather than as standalone equations to save space.
3）The paper reads more like a technical report, primarily focusing on implementation details while providing limited explanation of the underlying motivations and theoretical principles.

---

> ### Author Rebuttal · Authors · 2025-07-31
>
> Thank you for reviewing our paper. We appreciate your valuable feedback and will try to address your concerns below:
>
> **W1:** The paper's organization is somewhat disorganized. The core idea—selecting and aggregating key frames to balance long-term temporal modeling and computational efficiency could be presented more clearly. A top-down structure (general to specific) would improve readability, rather than introducing excessive details and equations in Section 3.1.
>
> **A1**: We thank you for the valuable comment. To improve clarity and follow a top-down structure, we will revise Section 3.1 to provide a clearer and more logically organized presentation of our method.
> In the revised Section 3.1 (Overview), we will remove non-essential mathematical equations and avoid excessive details,  reorganizing our approach.
>
> We begin by emphasizing the motivation behind our method:
> "Dynamic stereo matching aims to generate a sequence of temporally consistent disparity maps from stereo video inputs. While prior methods often struggle to capture long-range temporal dependencies efficiently, the proposed framework, PPMStereo, addresses this challenge by introducing a novel Pick-and-Play Memory (PPM) module. This module selectively aggregates high-quality reference frames into a compact memory buffer, enabling robust spatio-temporal modeling while maintaining computational efficiency.
> As illustrated in Fig. 2, PPMStereo is built on the DynamicStereo [22] backbone, with our PPM module serving as the core innovation."
>
> To improve the implementation details' readability, we will restructure the pipeline description as follows:
>
> (1) Feature Extraction
>
> (2) Cost Volume Construction
>
> (3) Context Encoding
>
> (4) Memory Buffer Initialization
>
> (5) Pick-and-Play Memory Module
>
> (6) Iterative Refinement
>
> If you still have concerns about our presentation framework, we would greatly appreciate any further feedback.
>
> **W2:** The excessive use of mathematical notation hinders readability. Additionally, some equations (e.g., (1) and (2)) are overly simplistic and could be presented inline rather than as standalone equations to save space.
>
> **A2:** We thank you for the constructive feedback. In the revised version, we will streamline the presentation by moving simpler expressions, such as Equations (1) and (2), to an inline format where appropriate.
> Additionally, we will revisit the mathematical content to remove non-essential notations and replace them with clearer explanations. We believe these revisions significantly enhance readability while preserving the depth and clarity of our core contributions.
>
> **W3:** The paper reads more like a technical report, primarily focusing on implementation details while providing limited explanation of the underlying motivations and theoretical principles.
>
> **A3:** We sincerely appreciate your thoughtful suggestions.  To further enhance the clarity of our paper, we will revise the abstract to more clearly state the research problem and the rationale behind our approach, and we will expand the introduction to deepen readers’ understanding of both our motivation and theoretical framework. Furthermore, we will reorganize Section 3.1 to highlight our motivation explicitly, and we will enrich Sections 3.2 and 3.3 with a more detailed discussion of the relevant theoretical principles. If you believe these revisions to the abstract, introduction, and Sections 3.1–3.3 remain insufficient, we would greatly appreciate any further feedback.
>
> **L1:** Does the keyframe extraction and compression process affect inference speed?
>
> **R1:** Thank you for the valuable comment. The keyframe extraction and compression process has a negligible impact on overall inference. The keyframe extraction and compression process corresponds to the "Pick Process", which is lightweight when evaluating reference frames. To validate this, we analyzed the "Pick Process" within the Pick-and-Play Memory mechanism over 20 iterations using 20-frame video sequences from the Sintel Stereo dataset (resolution: 1024 × 436) on an NVIDIA A100 GPU. The table below presents the average inference time for each component:
>
> |Model Variants|Inference Time (s)|
> |:---------------: | :--------------: |
> |Keyframe Extraction \& Compression Process (Pick Process) |11.8|
> |Random Keyframe Selection|11.4|
>
> As shown, the "Pick Process" introduces only a negligible increase in inference time, approximately 0.4 seconds, compared to a baseline that uses random keyframe selection. This marginal difference demonstrates that our keyframe extraction and compression pipeline incurs negligible computational overhead, preserving runtime efficiency while enabling reliable reference frame selection.
>
> **L2:** How many frames can the performance improvement from long-term temporal information be sustained? The paper only tests up to 20 frames?
>
> **R2:** We thank you for the insightful question. To ensure a fair comparison, our main experiments are conducted with 20-frame video sequences, following the protocol of prior work [20, 22]. Additionally, to assess performance on longer temporal sequences, we conduct experiments using varying sequence lengths, as shown below:
> |Number of Frames|Sintel Clean(TEPE)|Sintel Clean(Tδ1px)|Sintel Clean(Tδ3px)|Sintel Final (TEPE)|Sintel Final(Tδ1px)|Sintel Final(Tδ3px)|
> | :----------------: | :-----------------------: | :----------------: | :-----------------------: |:-----------------------: |:-----------------------: |:-----------------------: |
> |$T$ = 20|0.62|7.21|3.29|1.11|9.98|4.87|
> |$T$ = 30|0.61|7.17|3.29|1.09|9.92|4.81|
> |$T$ = 40|0.61|7.17|3.28|1.09|9.93|4.82|
> |$T$ = 50|0.61|7.18|3.28|1.09|9.92|4.82|
>
> As the sequence length increases, performance initially improves, but eventually reaches a saturation point. This saturation arises because overly long sequences introduce redundant or irrelevant information, which eventually contributes little to disparity estimation.
> Overall, the performance improvement of PPMStereo from long-term temporal information can be sustained up to 30 frames without degradation.

---

> > ### Comment · Reviewer_HYhV · 2025-08-07
> > **Weakness addressed**
> >
> > My main concerns were the paper's structure and the computational cost and improvement range of the method. The author's response has effectively addressed my concerns, so I have increased my score.

---

> > > ### Author Response · Authors · 2025-08-07
> > >
> > > We sincerely thank the kind reviewer for their thoughtful feedback and valuable suggestions, which have greatly contributed to improving the quality of our manuscript.

---

### Comment · Area_Chair_s3vi · 2025-08-06
**Re-Rebuttal**

Dear Reviewers,

Thank you to those who have shared your thoughts regarding authors' rebuttal.

Please also note that reviewer cannot just click the acknowledgement without providing feedback/comments. Your comments or reply to authors are compulsory based on the NeurIPS' rule for reviewers this year.

Your engagement with authors and other reviewers is highly appreciated.

Your AC

---

### Decision · Program_Chairs · 2025-09-17

**Decision:**

Accept (poster)

**Comment:**

This paper received positive ratings 1) Borderline Accept and (2) Accept from reviewers.
The main contribution of the paper is the introduction of the pick-and-play memory mechanism which can model long-range temporal consistency leading to strong experimental performance for stereo matching in stereo video sequence.

While reviewers have concerns regarding 1) lack validation of number of frames being able to be handled by the method, 2) lack discussions of computational cost, 3) lack discussion of hyperparameters.

Authors provided detailed response in the rebuttal. All reviewers acknowledge that most of their concerns are addressed and would recommend accept.

AC concurs with reviewers’ comments and would recommend Accept.